# Metal Deposition Induced by the Step Region of Si (111)-(7 × 7) Surface

**Wenxin Li [1,2], Wanyu Ding [3]** 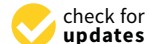 **, Youping Gong [1]** **and Dongying Ju [2,4],***

1   College of Mechanical Engineering, Hangzhou Dianzi University, Hangzhou 310018, China; wwenxindiaolong@hdu.edu.cn (W.L.); gyp@hdu.edu.cn (Y.G.)
2   Department of High-Tech Research Center, Saitama Institute of Technology, Fusaiji 1690, Fukaya 369-0293, Japan
3   School of Material Science and Engineering, Dalian Jiaotong University, No.794 Huanghe Road, Shahekou District, Dalian 116028, China; dwysd@djtu.edu.cn
4   Ningbo Haizhi Institute of Materials Industry Innovation, Ningbo 315000, China
*   Correspondence: dyju@sit.ac.jp

**Abstract:** Scanning tunneling microscope results showed that Au and Fe atoms were steamed on the Si (111)-(7 × 7) substrate surface, with or without the step region. The experimental comparison proved that the induced effect of the step region is a controllable process, which $CH_3OH$ can adjust. In this paper, the latest progress on the dynamic phenomenon on the step region is discussed, including three deposition types: strong deposition, weak deposition, and the new quasi deposition. With a relatively weak interaction between Au and Si atoms, the linearity of the weak deposition is present in the step region. In contrast, Fe atoms tend to form a strong deposition along the boundary line between the flat and step regions. Different depositions correspond to different surface potential energy: a newly formed surface is stabilized by a quasi-potential made by breaking, and a metal atomic structure can be stabilized by forming several quasi depositions. After discussing the good adsorption properties, $CH_3OH$ can be used as an intermediate layer on the step region. As an important result of quasi deposition, a regular linear Fe cluster structure is created, which is perpendicular to the boundary line.

**Keywords:** $CH_3OH$; step region; quasi deposition; linear clusters structure

## 1. Introduction

With the progression of atomic-level testing technology [1,2], a series of interesting dynamic phenomena [3,4] on the solid surface, such as the formation of typical structures and fluctuation of the step region [5,6], have been found. The important substrate material, Si (111)-(7 × 7), is a reconstructed surface formed with many regular triangle structures in a horizontal directin. In a vertical direction, each triangle structure can be divided into a top layer and a rest layer, which have fixed adsorption/deposition sites. Among them, those on the top layer are called corner sites and center sites, and those on the rest layer are called rest sites. Its atomic structure growth is significant to the study of metal deposition [7,8]. However, despite such theoretical and functional values, there are a very limited number of application cases that can exactly reach the atomic level. This is partially due to the difficulties in obtaining stable, regular, and high-density atomic structures. Additionally, as the size of the metal deposition decreases to a certain extent (around 10 nm), some unique phenomena follow, such as the small-size effect, surface effect, and so on [9,10].

In recent years, researchers have noted that metal atoms on the step region are particularly active. As the non-equilibrium effect of the silicon substrate is eliminated, the surface is deformed and the flat surface turns into a step region with a height difference. Intensive theoretical and experimental research has been performed on exploring new behaviors in the step region. Since they are not completely surrounded by Si atoms,

metal atoms are more likely to interact with each other, as well as with the environment, showing an induction process in the step region. Increasingly, the findings show that the behavior of metal atoms in the step region of the Si (111)-(7 × 7) surface is different from that on the flat region, which tends to achieve typical atomic structures [11].

The interaction between the step region and the metal atoms is induced by its elastic fields. The structure induction contributes to the surface potential energy, thus affecting the atomic structures as well as their distribution [12]. Accordingly, the shape and size of various metal depositions can be adjusted and sometimes undergo reversible characteristic changes in response to variations in the interaction, which can induce changes in their reactivity and selectivity. With the help of a scanning tunneling microscopy (STM) system and first-principles calculation, adsorption/deposition sites on Si (111)-(7 × 7) surface have been systematically investigated and identified by our group [13–15]. Au and Fe atoms were scanned and analyzed, and some phenomena different from the flat region were found at the atomic-level. In our study, the structure induction changed the surface potential energy to a certain extent, thus affecting the interaction between the metal deposition and substrate surface.

Typical models of strong and weak deposition have been proposed. The interaction between the Au and Si atoms is presented as a weak interaction, forming a weak deposition. In contrast, Fe deposition is regarded as a strong type, which can easily form Fe–Si and Fe–Fe bonds [11,16]. After a series of classification models were created, the focus became how the atomic structure would respond to surface potential energy changes, especially in the induction process. Additionally, to adjust the induced effect influenced by the step region properly, $CH_3OH$ was selected as an intermediate layer [16–18]. From a macro perspective, the step region or $CH_3OH$ adsorption could be explained as an amplification/reduction effect between metal and Si atoms [19]. Recent studies have shown that some regular atomic structures can be achieved if the surface potential energy is adjusted instead of the material composition [20,21].

## 2. Materials and Methods

$CH_3OH$, Au, and Fe atoms were adsorbed, streamed, and observed by a scanning tunneling microscopy (STM) system (JEOL Ltd., Akishima-shi, Japan). As shown in Figure 1, STM experiments were implemented in the observation chamber with a base pressure of $1 \times 10^{-8}$ Pa, while the vacuum degree was below $4 \times 10^{-8}$ Pa in the treatment chamber. The treatment chamber was equipped with 4 holders, which could be used to handle four samples simultaneously. In our experiments, the specific procedure was as follows.

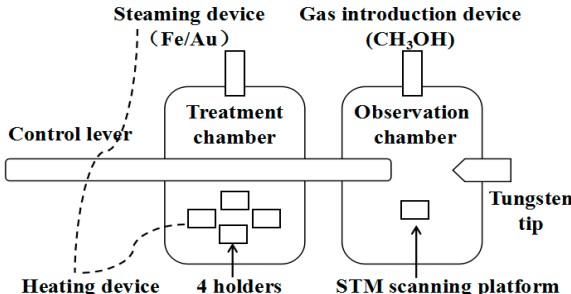

**Figure 1.** Structure diagram of the experimental platform.

First, several n type Si (111) substrates were ultrasonically precleaned in acetone, ethanol, and deionized water, separately. Substrates were degassed in the treatment chamber at 400 °C. Then, the sample was repeatedly heated at 1200 °C until a clean, well-ordered Si (111)-(7 × 7) reconstructed surface was obtained. With an increase in the high-temperature flash frequency, the short-term thermal imbalance made it easy to form a step region on the substrate surface. These samples were then sent into the observation chamber, which was filled with $CH_3OH$. The concentration of $CH_3OH$ was kept at $10^{-6}$ Pa

for 30–45 s. During the adsorption process, gaseous ion composition was detected in the observation chamber. After we explored and summarized the $CH_3OH$ adsorption rulers, Si (111)-(7 × 7)-$CH_3OH$ samples were moved back to the treatment chamber. Using the steaming device in STM system, metal atoms were separately deposited on the samples of Si (111)-(7 × 7) and Si (111)-(7 × 7)-$CH_3OH$. The deposition process could be controlled by adjusting the steaming time and temperature of the heating device (Figure 1). After adjusting the position of the holders, the metal atoms were steamed by heating a W filament (>1000 °C) with Au or Fe wire (purity > 99.995%). Unlike the size (0.2 × 15 × 110 mm$^3$ previously used in reference [11], we chose a size of 0.3 × 15 × 110 mm$^3$. The improved steaming device not only provided a higher steaming temperature, but also reduced the influence on metal atomic structures (especially the linear structure).

The tungsten tip was made using an electrochemical etching, and STM images were obtained in a constant current mode at room temperature. Considering the complexity of the multilayer structure, when the deposition was scanned, STM would also convert it into 3D images. The specific measurement of morphology not only provided compositional data of the atomic structures, but also helped us to distinguish and establish models of various metal deposition. Additionally, the experiments focused on the chemical stability of Fe quasi-deposition on the surface of Si (111)-(7 × 7)-$CH_3OH$. Before and after being exposed to thin air, the samples were translated into a composition test chamber, separately, where the sample was tested in situ by the GammadataScienta SES-100 X-ray photoelectron spectroscopy (XPS) system (Pleasanton, CA, USA). With the aim of improving the signal-to-noise ratio of the data, the area of XPS measurement was kept as 10 × 10 μm$^2$ for all tests.

## 3. Results

### 3.1. Metal Deposition on the Step Region

Based on a low-dimensional structure, the growth mechanism of the metal deposition has become increasingly important. STM and other advanced equipment have increased our observation accuracy to the atomic level. However, the functional requirements of the metal deposition have increased, such as regular distribution, efficiency reaction, and so on. For relatively lower deposition concentrations, a structure-induction process was proposed to achieve high-density atomic structures. Thermal annealing of the Si (111)-(7 × 7) surface in ultra-high vacuum conditions produced the step region separated by atomically smooth singular terraces. As an important micro-phenomenon, the step region can not only change the anisotropy of the surface force field, but also plays an important role in the surface electron density. The formation of the step region is the result of a gradual stretching of the marginal parts. The degree of tension can be reflected in the height difference, as shown in Figure 2. When the height difference is relatively small, a cliff area is formed, which is defined as the primary stage (Figure 2a). With the expansion of the step region, a stable basin area is formed, as shown in Figure 2b.

Preferentially, metal atoms were directly deposited on the Si (111)-(7 × 7) surface, where the atomic structure could be observed in a relatively low concentration. In this study, Au and Fe atoms were taken as two typical deposition types to scan and compare. On the flat region of the Si (111)-(7 × 7) surface, Au atoms were deposited on the deeper position (Figure 3) without forming an Au–Si bond. The interaction among Au atoms was also weak, showing uniform distribution in Figure 4a. The new phenomenon on the step region (Figure 4b) was significantly different from this situation, which showed that most Au atoms began to form a linear structure. Moreover, as the deposition time increased, more linear atomic structures that were perpendicular to the boundary line were found. With the growth in each linear structure, although some scattered Au atoms were also found, the number of which was relatively small. Through these interesting phenomena, it can be inferred that the interaction among the Au atoms was induced on the step region of the Si (111)-(7 × 7) surface. In contrast to the Au atoms, the Fe atoms were usually deposited on the corner/center sites of the Si (111)-(7 × 7) surface (Figure 3). In Figure 5a,

several Fe cluster structures are randomly formed on the flat region of the Si (111)-(7 × 7) surface, forming both Fe–Si and Fe–Fe bonds. In Figure 5b, Fe atoms begin to deposit along the boundary line of the flat and step region. At the same deposition condition, the size of the Fe cluster structure on the step region was larger than that of the flat region (Figure 5c). Once again, this typical phenomenon verified the induced effect of the step region. By continuing to increase the deposition time (Figure 5d), more Fe cluster structures were found along the boundary line.

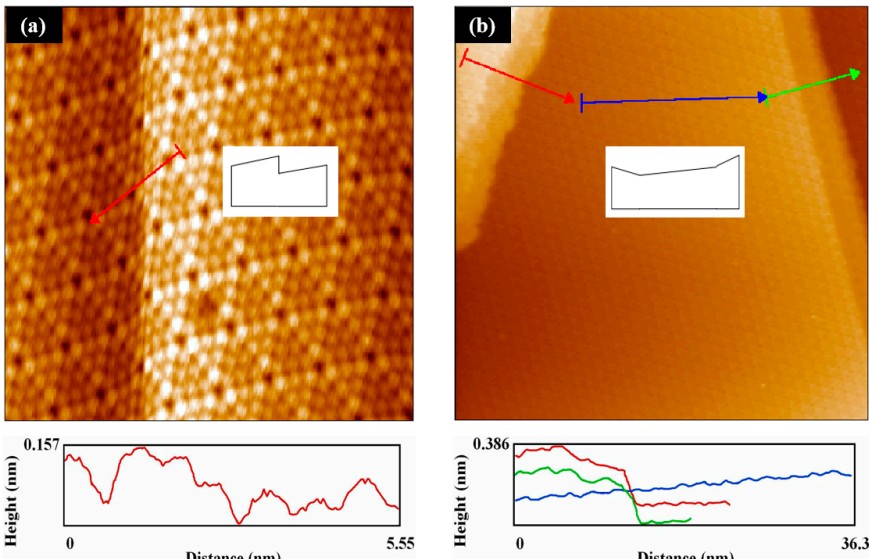

**Figure 2.** Scanning tunneling microscopy images of the step region on the Si (111)-(7 × 7) reconstructed surface. Each measurement is marked with different colors (red, green, and blue): (**a**) a cliff structure of the step region, the height difference of which is below 0.1 nm; (**b**) a basin structure of the step region, the height difference of which is more than 0.2 nm.

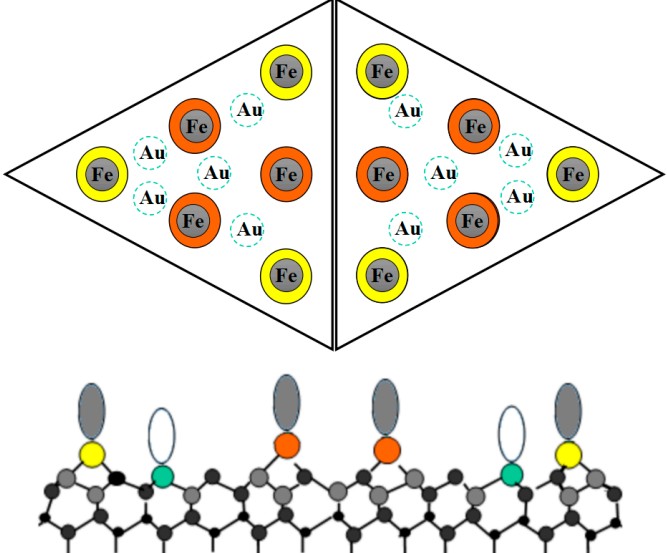

**Figure 3.** The deposition model of metal atoms on the Si (111)-(7 × 7) surface. The Fe and Au atoms were deposited on different Si (111)-(7 × 7) surfaces, separately. The Fe atoms tended to react with the Si atoms on the top layer, forming Fe–Si bonds; Au atoms were deposited on deeper sites without reacting with Si atoms. Among them, corner sites are marked as yellow, center sites are marked as orange, and rest sites are marked as green.

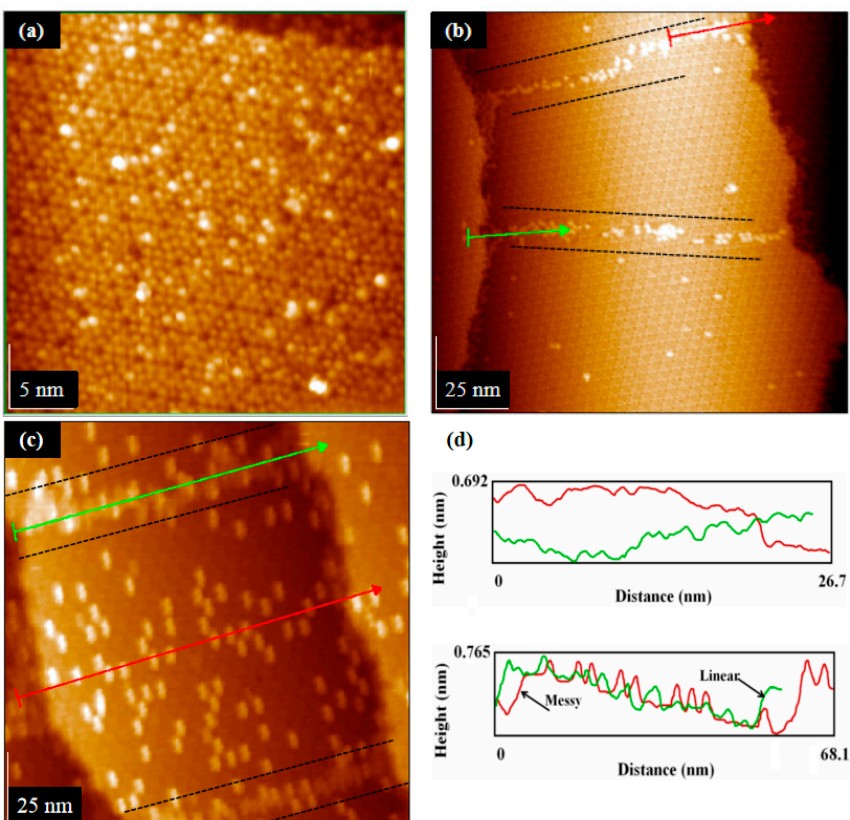

**Figure 4.** Scanning tunneling microscopy images of (**a**) the flat region of the Si (111)-(7 × 7) surface, which was steamed with Au atoms, $10^{-6}$ Pa, 20 s; (**b**) the step region of the Si (111)-(7 × 7) surface, which was steamed with Au atoms, $10^{-6}$ Pa, 20 s; (**c**) the step region of the Si (111)-(7 × 7) surface, which was steamed with Au atoms, $10^{-6}$ Pa, 30 s. (**d**) is the height measurement of (**b**) and (**c**): two Au linear structures were measured in (**b**), and some messy Au atoms (red) were also measured to compare with the Au linear structure (green).

## *3.2. Classification Model and CH$_3$OH Adsorption*

### 3.2.1. Deduction of the Theoretical Prediction

The formation of the atomic structure on the step region was a special case; if metal atoms showed a characteristic on a flat surface, the characteristic could be induced on the step region. Different metal depositions corresponded to different interactions, which were reflected in the forming of a different atomic structure. Even on deeper deposition sites, Au atoms were hardly affected by the Si atoms, showing a relatively uniform regular distribution on the flat region, while the Fe deposition easily formed a cluster structure under the double influence of the interaction and surface reaction. These typical atomic structures were effectively affected by structural induction, which was more obvious on the step region. In reference to the formation of chemical bond, the metal deposition can be divided into two types on the step region of the Si (111)-(7 × 7) surface (Figure 6a,b). Au and Fe corresponded to weak deposition and strong deposition, respectively. Under the effect of structure induction, Au atoms began to form a regular linear structure, while the cluster structure of Fe deposition was further enlarged. When considering the development of functional materials, the single atom was far from application, and both regular linearity and clustering character were essential. With the weakening of linear characteristics, the formation probability of cluster structure increased. Notably, the large cluster structure size was not conducive to high reaction efficiency and functional density. Thus, between the strong and weak deposition, there exists a transition deposition. There are enough reasons to think that the regular structure can be found if the ladder induction is adjusted in a suitable range. As a theoretical prediction, a new model is proposed in Figure 6, namely,

quasi deposition. It can be inferred that the new quasi deposition not only has the advantage of linearity but also keeps the formation of the cluster structure. In this way, the introduction of an intermediate layer, which could adjust the induced effect, has become the focus of our further study.

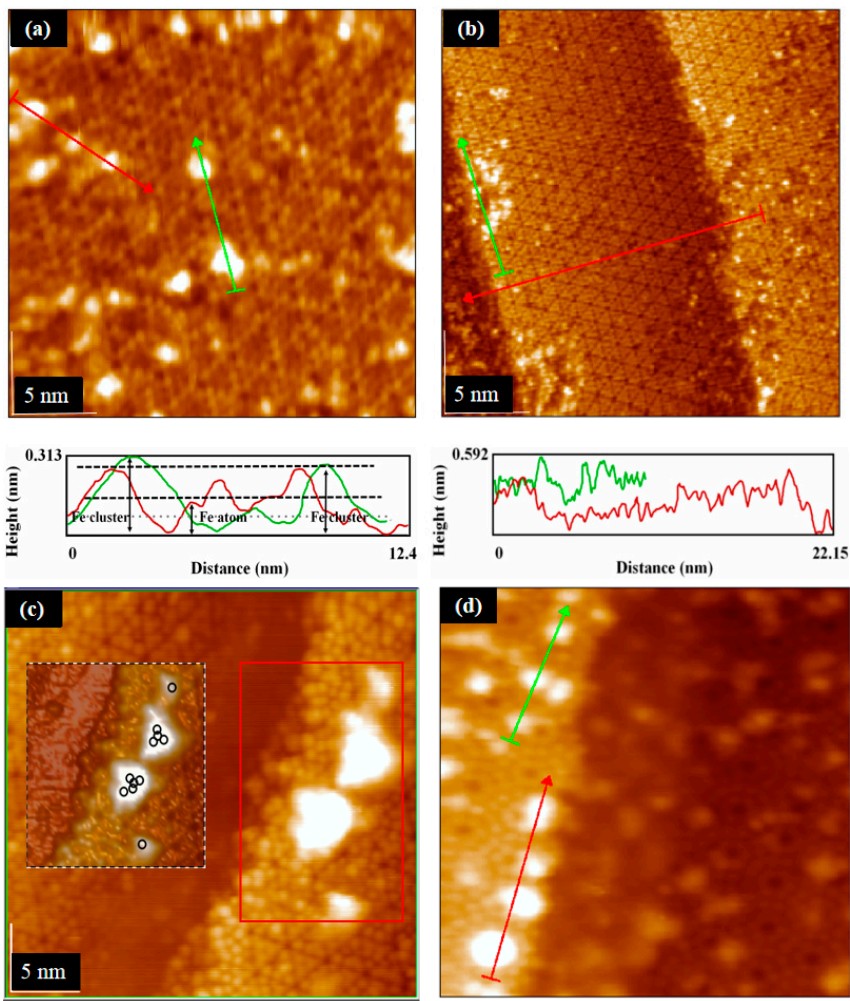

**Figure 5.** Scanning tunneling microscopy images of (**a**) the flat region of the Si (111)-(7 × 7) surface, which was steamed with Fe atoms, $10^{-6}$ Pa, 20 s; (**b**) the step region of the Si (111)-(7 × 7) surface, which was steamed with Fe atoms, $10^{-6}$ Pa, 15 s; (**c**) the step region of the Si (111)-(7 × 7) surface, which was steamed with Fe atoms, $10^{-6}$ Pa, 20 s, showing a larger cluster structure; (**d**) the step region of the Si (111)-(7 × 7) surface, which was steamed with Fe atoms, $10^{-6}$ Pa, 30 s.

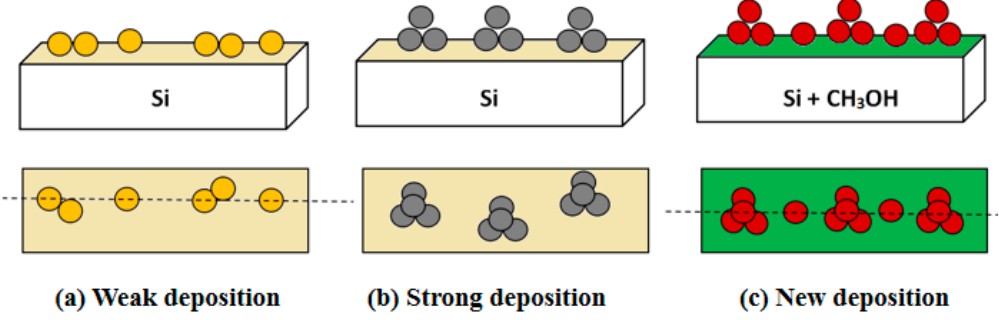

**Figure 6.** Three typical metal deposition models: (**a**) the yellow one represents weak deposition, which tends to form a linear structure; (**b**) the gray one represents a strong deposition, which easily forms a cluster structure; (**c**) the red represents a new type of deposition, which could form linear clusters.

### 3.2.2. CH$_3$OH Adsorption on the Step Region

Through long-term research, CH$_3$OH was selected as the suitable intermediate layer, which can be adsorbed on the step region before the metal deposition. Using the detection data of a mass spectrometer, the composition of the gas ion in the observing chamber is shown in Figure 7a. A dissociation reaction was found during the adsorption process of CH$_3$OH gas:

$$CH_3OH = CH_3O^- + H^+ \tag{1}$$

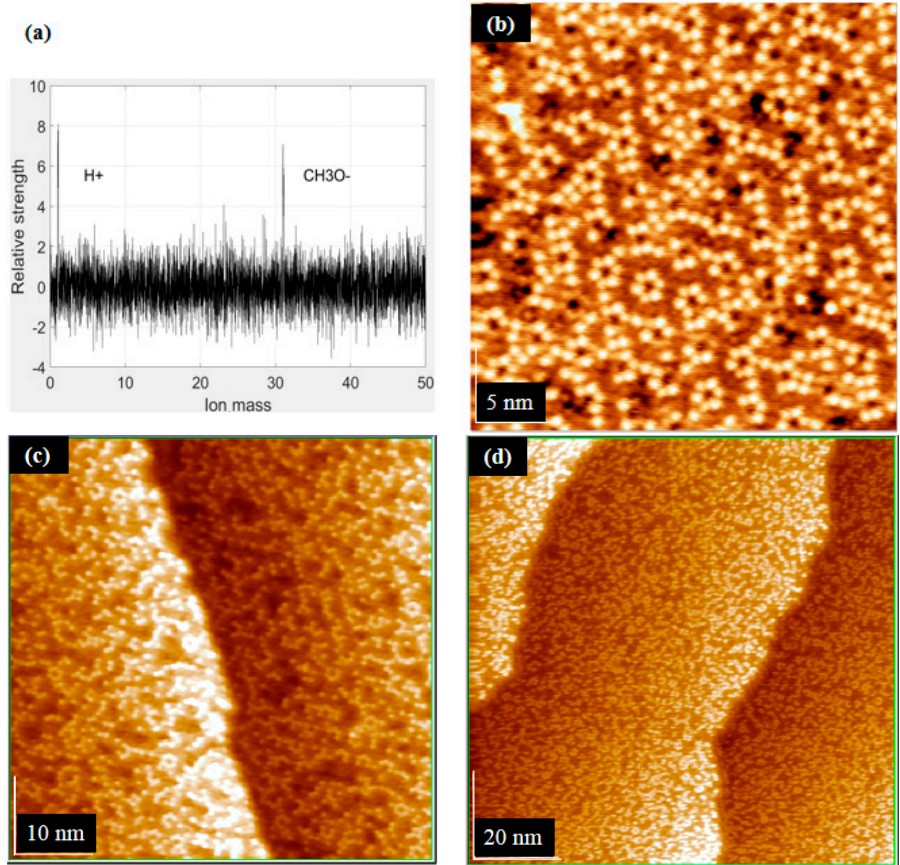

**Figure 7.** (**a**) Mass spectra image of CH$_3$OH adsorption. STM images of (**b**) the flat region of the Si (111)-(7 × 7) surface with CH$_3$OH concentration of $10^{-6}$ Pa, 30 s; (**c**) the step region of the Si (111)-(7 × 7) surface with CH$_3$OH concentration of $10^{-6}$ Pa, 30 s; (**d**) the step region of the Si (111)-(7 × 7) surface with CH$_3$OH concentration of $10^{-6}$ Pa, 45 s.

The CH$_3$OH molecule dissociated on the Si (111)-(7 × 7) surface by forming Si–OCH$_3$ and Si–H. Whether on the flat or step region, CH$_3$OH adsorption presented a relatively uniform distribution, as shown in Figure 7b,c. When the adsorption time reached about 30 s, the coverage rate was maintained at around 50%, which left enough sites for forming a linear or cluster structure. As the adsorption time continued to increase (Figure 7d), methanol coverage increased briefly but quickly returned to a stable level of 50%. Additionally, the local electronic state may change with the dynamic motion of CH$_3$O$^-$/H$^+$ in adjacent half-unit cells, which also leads to the fluctuation of a potential barrier. With the change in the surface potential energy, the interaction of the metal deposition also changed, resulting in new deposition types and atomic structures. All these findings imply that CH$_3$OH can be used as an ideal intermediate layer in the structure induction process.

### 3.3. The Realization of Quasi Deposition

Au and Fe atoms were deposited on the new Si (111)-(7 × 7)-CH$_3$OH surface, separately. According to the theory of surface potential energy, the interaction between atoms is the key to the shape, size, and distribution of metal deposition. Under the same deposition condition in Figure 4b, the linear structure could not be found on the step region of the Si (111)-(7 × 7)-CH$_3$OH surface (Figure 8a). It can be seen that CH$_3$OH played a reduced role in the structure induction process of the step region. In contrast to Figure 4, CH$_3$OH also limited the growth of Fe clusters on the step region, the size of which decreased (Figure 8b). The smaller the cluster structure, the higher reaction efficiency and functional density. In Figure 8c, a linear structure is successfully formed among each Fe cluster. On the one hand, the new Fe deposition is no longer confined to the boundary line, and grows in the vertical direction like the Au atomic structure of the weak deposition. On the other hand, although the size of the Fe cluster structure is reduced to the level of the flat region, the linearity is significantly improved, indicating the essential difference to the strong deposition before. Based on the new model in Figure 6, further derivation could occur according to the actual experimental results.

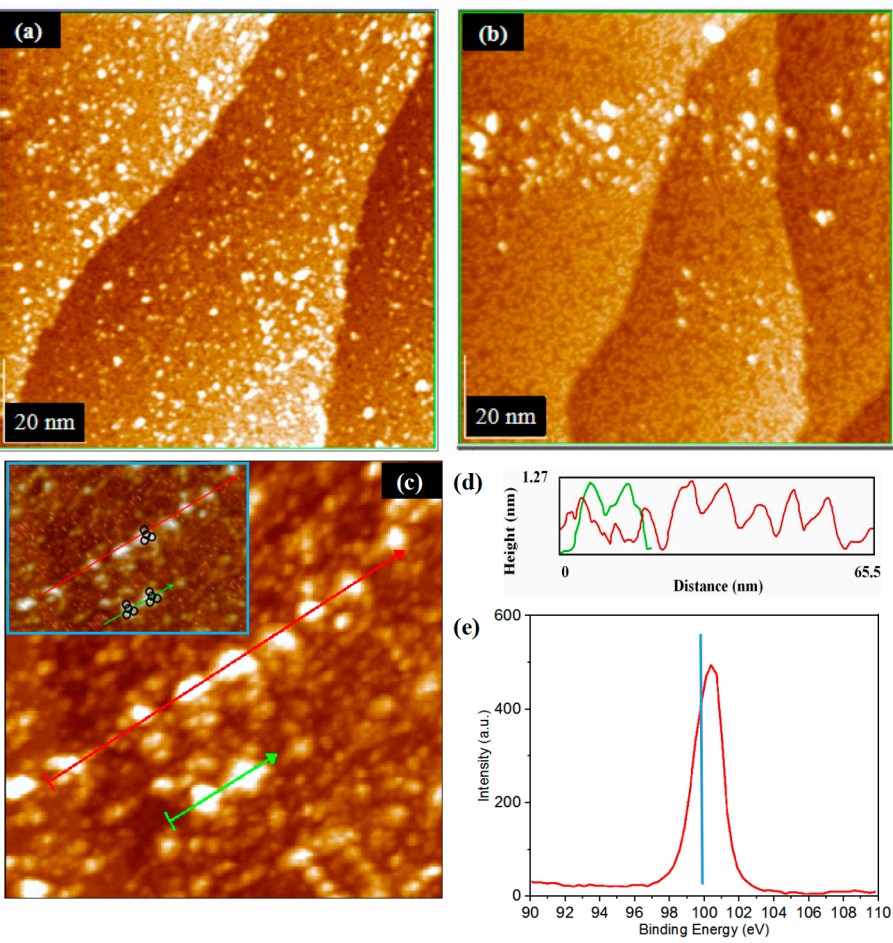

**Figure 8.** STM images of (**a**) the step region of the Si (111)-(7 × 7)-CH$_3$OH surface, which is steamed with Au atoms, $10^{-6}$ Pa, 20 s; (**b**) the step region of the Si (111)-(7 × 7)-CH$_3$OH surface, which is steamed with Fe atoms, $10^{-6}$ Pa, 20 s; (**c**) the step region of the Si (111)-(7 × 7)-CH$_3$OH surface, which is steamed with Fe atoms, $10^{-6}$ Pa, 15 s; (**d**) The height measurement of (**c**); some messy Fe clusters (green) were also measured to compare with the Fe linear clusters (red); (**e**) XPS spectra of (**b**): the peak represents the Si–Si bond, and no Si–Fe bond was found.

When a metal deposition takes the form of single atoms, it is difficult to form a regular structure due to the weak surface energy on the flat region of the Si (111)-(7 × 7) surface.

An induction process is needed to improve the atomic interaction in Au deposition, while the use of $CH_3OH$ would reduce the amplification effect of the step region. When a metal deposition takes the form of several clusters, the key to forming a linear clusters structure is to reduce the strong surface energy among each atom while avoiding adverse effects, such as small-size effects and surface effects. It was proven that $CH_3OH$ could effectively reduce the size of the Fe cluster structure on the step region, thus significantly increasing the functional density of atomic-level material. Through the 3D method of the STM system (Figure 8c), an atomic stacking structure of linear Fe clusters can be observed clearly. No obvious height differences were found between each cluster. Evidently, this new quasi-atomic structure has the dual characteristics of strong and weak deposition. Additionally, our new quasi deposition has more specific mechanisms to explore than the simple increase and decrease in the surface energy among the metal and Si atoms. For example, XPS was used to better investigate the surface potential energy of Si (111)-(7 × 7)-$CH_3OH$-Fe. As a result, no Si–Fe bond was found. The XPS spectra are shown in Figure 8d, proving the existence of a pure Fe atomic structure. These investigations showed that, to a certain extent, there is a promotion to form high-density functional material on the restructured Si (111)-(7 × 7)-$CH_3OH$ surface, especially according to the coating and deposition technology we adjusted.

## 4. Conclusions

In this study, where we examined a series of interesting phenomena on the flat and stepped region, the law of metal deposition was gradually discovered and deduced. Influenced by structure induction, Au and Fe atoms exhibited more active atomic behavior. The metal deposition on the step region of the Si (111)-(7 × 7) surface is a typical case [22], that is, if the interaction among the metal and Si atoms involves an induction process, the formation of a metal atomic structure is difficult to explain without considering the change in surface potential energy [23–25]. Accordingly, the theory of strong/weak interaction was put forward on the step region of the Si (111)-(7 × 7) surface so that the feature of prominent induction and/or co-induction could be reflected in a prerequisite state for the metal deposition. On the basis of the classification model, $CH_3OH$ was further used to adjust the induction process. On the step region of Si (111)-(7 × 7)-$CH_3OH$, metal atoms exhibited different atomic structures, as well as distributions. The experimental results are in agreement with our theoretical prediction, especially the successful formation of a new quasi-deposition. These prerequisites states (structure induction and gas adsorption) have considerable potential for precisely controlling the atomic structure formation, and will help further studies of function materials on the substrate surface concerning the realization of atomic-level devices [26,27].

**Author Contributions:** W.L. and W.D. grew the samples and performed the STM, XPS, and mass spectrometer measurements. W.L. and Y.G. wrote and analyzed the data and wrote the manuscript. W.D. and D.J revised the manuscript. D.J. supervised the work and offered suggestions. All authors read and approved the final manuscript.

**Funding:** This work was supported by the Nano Project of Saitama Institute of Technology in Japan, the National Natural Science Foundation of China (No. 51472039 and 51772038), and Natural Science Foundation of Liaoning Province, China (No. 201202024).

**Acknowledgments:** This work was supported by the Project of Advanced Science Research Laboratory of Saitama Institute of Technology. The author thanks Tanaka for the discussion of the issues. Additionally, thanks for the XPS testing provided by Institute for Solid State Physics of the University of Tokyo, and Komori and Iimori for the experiments.

**Conflicts of Interest:** The authors declare no conflict of interest.

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
