# Peer review of "Metal Deposition Induced by the Step Region of Si (111)-(7 × 7) Surface"

_coatings, doi:10.3390/coatings11030281_

Round 1

Reviewer 1 Report

Dear Author(s),
The quasi deposition method is quite impressive, and thanks for quite work on this subject. I do have some questions for the authors.
1- What do you mean with 7x7 on Si(111)? It will be better to define it in the introduction.
2-Authors explained how to prepare the samples. Please add an illustration of how samples are prepared during deposition. The explanation of the sample preparation is quite confusing. The reader will understand quickly with the support of the illustration.
3- In Figure 2, the bottom illustration is not clear. Please name each colored object in the illustration. 
4- I could not find the definition of the weak and strong deposition until Figure 5. The definition of each deposition would be better defined in the introduction part. 

Also, the manuscript needs attention grammatically. 

Thanks

Author Response

Thank you very much, please check the attachment.

Reviewer 2 Report

The present paper is very interesting. It can be accepted after minor revision.

I have identified the followings:

  1. in few places, the 3 from CH3 has to be to subscript.
  2. the caption figure of Figure 7 has to be imporved: it shall be reduced to a relevant sentance and the numbers shall be standing for the several subfigures.

Author Response

(The authors gave the same response as above.)

Reviewer 3 Report

The paper entitled “Study on metal deposition induced by the step region of Si 3 (111)-7×7 surface” focuses on the deposition sites of Au and Fe atoms on Si (111)-7×7 surface by using a scanning tunneling microscope (STM). The authors examine the structure induction changes that affect the interaction in metal deposition and how the atomic structure responds to surface potential energy changes, especially in the induction process. This fundamental paper should be interesting from a scientific and practical point of view because it proposes a new model of typical metal deposition which could form linear clusters.

I would like to recommend the publication of the manuscript in this journal after fulfilling the following recommendations:

  1. Please, avoid using abbreviations in the abstract;
  2. The aim of the study should be precisely formulated;
  3. Superscribes and subscripts in the text should be carefully revised;
  4. It is not clear from the “Materials and Methods” section how the relative strength of CH3OH adsorption on Si substrate was measured.
  5. It is not clear from the text and figures how Fe and Au atoms are distinguished from one another;
  6. The authors should indicate with higher precision which lines used for indication of the height difference of the samples correspond to a particular image.
  7. The title of section 4 is not suitable;
  8. In the last section, the authors said that “The experimental result is in good agreement with our theoretical prediction, especially the successful formation of a new quasi deposition.”. It would be better for authors to give special focus on that “theoretical prediction” as making it part of a particular section or sub-section in the text;
  9. In some places, the English style and grammar have to be improved. Some examples of inadequate style: “While Fe atoms are more likely to form strong deposition, along the boundary line between 19 flat and step regions”; “After the classification model was speculated, the question again rises to how the atomic structure responds to surface potential energy changes, especially in the induction process.”; “Continue to increase the deposition time (Fig. 4 d), more and more Fe cluster structures were found along the boundary line.”; “Aim to improve the signal-to-noise ratio of data, the area of XPS measurement was kept as 100 μm in diameter for all tests.”; “The degree of tension is reflected by the height difference, just as the Figure 1 shown.”; …and many more…

Author Response

(The authors gave the same response as above.)

Reviewer 4 Report

In this work, the influence of metal deposition induced by the step region of the Si (111)-7×7 surface has been studied. Although the topic of the work may be of interest, the authors do not make clear the field of application of the work and the paper presents aspects that need to be improved. Therefore, I recommend a very important revision.

Summary

-The novelty of the subject matter of this paper is not clearly reflected.

Introduction

-Could you indicate the objectives of the paper?

Material and methodology

-In the methodology it is mentioned about samples, could you please indicate the number of samples analysed during the experiment?

-On line 90 it says: "Unlike the size (0.2×15×110 mm3) used previously in reference [11], we chose the size as 0.3×15×110 mm3." This can be justified.

-It would be possible to put "mm3" as "mm3".

-On line 94 it says: "STM images were obtained in a constant current mode at room temperature". Are these values known?

-I recommend some images of the experimental development.

Results

-On lines 174-178: A structural analysis as a function of time is mentioned, this methodology is not clearly stated in the methodology.

-In the results, the discussion of the results is missing.

-Equation (1) is mentioned twice.

-Why have all the images been grouped together instead of appearing when referenced?

Conclusions

-The conclusion section has been replaced by "4.results", I suggest it be changed.

-I suggest that references and discussions be developed throughout the result section.

Author Response

(The authors gave the same response as above.)

Round 2

Reviewer 3 Report

Тhe paper may be accepted in its present form.

Reviewer 4 Report

In this work, the influence of metal deposition induced by the step region of the Si (111)-7×7 surface has been studied. The changes made to the paper have included the recommendations made by the reviewers. I congratulate the authors for their work.